# ON THE EFFECT OF THE ACTIVATION FUNCTION ON THE DISTRIBUTION OF HIDDEN NODES IN A DEEP NETWORK

## ABSTRACT

We analyze the joint probability distribution on the lengths of the vectors of hidden variables in different layers of a fully connected deep network, when the weights and biases are chosen randomly according to Gaussian distributions, and the input is in $\{-1, 1\}^N$. We show that, if the activation function $\phi$ satisfies a minimal set of assumptions, satisfied by all activation functions that we know that are used in practice, then, as the width of the network gets large, the "length process" converges in probability to a length map that is determined as a simple function of the variances of the random weights and biases, and the activation function $\phi$.

We also show that this convergence may fail for $\phi$ that violate our assumptions.

## 1 INTRODUCTION

The size of the weights of a deep network must be managed delicately. If they are too large, signals blow up as they travel through the network, leading to numerical problems, and if they are too small, the signals fade away. The practical state of the art in deep learning made a significant step forward due to schemes for initializing the weights that aimed in different ways at maintaining roughly the same scale for the hidden variables before and after a layer [9, 4]. Later work [7, 14, 2] took into account the effect of the non-linearities on the length dynamics of a deep network, informing initialization policies in a more refined way.

In this paper, we continue this line of work, theoretically analyzing what might be called the "length process". That is, for a given input, chosen for simplicity from $\{-1, 1\}^N$, we study the probability distribution over the lengths of the vectors of hidden variables, when the parameters of a deep network are chosen randomly. We analyze the case of fully connected networks, with the same activation function $\phi$ at each hidden node and $N$ hidden variables in each layer. As in [14], we consider the case where weights between nodes are chosen from a zero-mean Gaussian with variance $\sigma_w^2/N$, and where the biases are chosen from a zero-mean distribution with variance $\sigma_b^2$.

Our first result holds for activation functions $\phi$ that satisfy the following properties: (a) the restriction of $\phi$ to any finite interval is bounded; (b) as $z$ gets large, $|\phi(z)| = \exp(o(z^2))$; (c) $\phi$ is measurable. We refer to such $\phi$ as *permissible*. Note that conditions (a) and (c) both hold for any non-decreasing $\phi$.

We show that, for all permissible $\phi$ and all $\sigma_w$ and $\sigma_b$, as $N$ gets large, the length process converges in probability to a length map that is a simple function of $\phi$, $\sigma_w$ and $\sigma_b$. This length map was first discovered in [14], where it was claimed that it holds for all $\phi$; it has since been used in a number of other papers [15, 17, 12, 10, 16, 1, 13, 5].

In Section 4, to motivate our new analysis, we provide examples of $\phi$ that are not permissible that lead the length processes with arguably surprising properties. For example, we show that, for arbitrarily small positive $\sigma_w$, even if $\sigma_b = 0$, for $\phi(z) = 1/z$, the distribution of values of each of the hidden nodes in the second layer diverges as $N$ gets large. For finite $N$, each node has a Cauchy distribution, which already has infinite variance, and as $N$ gets large, the scale parameter of the Cauchy distribution gets larger, leading to divergence. We also show that the hidden variables in the

second layer may not be independent, even for some permissible $\phi$ like the ReLU. The results of this section contradict claims made in [14, 10].

Section 5 describes some simulation experiments verifying some of the findings of the paper, and illustrating the dependence among the values of the hidden nodes.

Our analysis of the convergence of the length map borrows ideas from Daniely, et al. [2], who studied the properties of the mapping from inputs to hidden representations resulting from random Gaussian initialization. Their theory applies in the case of activation functions with certain smoothness properties, and to a wide variety of architectures. Our analysis treats a wider variety of values of $\sigma_w$ and $\sigma_b$, and uses weaker assumptions on $\phi$.

## 2 PRELIMINARIES

### 2.1 NOTATION

For $n \in \mathbb{N}$, we use $[n]$ to denote the set $\{1, 2, \ldots, n\}$. If $T$ is a $n \times m \times p$ tensor, then, for $i \in [n]$, let $T_{i,:,:} = \langle T_{i,j,k} \rangle_{jk}$, and define $T_{i,j,:}$, etc., analogously.

### 2.2 THE FINITE CASE

Consider a deep fully connected width-$N$ network with $D$ layers. Let $W \in \mathbf{R}^{D \times N \times N}$. An activation function $\phi$ maps $\mathbf{R}$ to $\mathbf{R}$; we will also use $\phi$ to denote the function from $\mathbf{R}^N$ to $\mathbf{R}^N$ obtained by applying $\phi$ componentwise. Computation of the neural activity vectors $x_{0,:}, ..., x_{D,:} \in \mathbf{R}^N$ and preactivations $h_{1,:}, ..., h_{D,:} \in \mathbf{R}^N$ proceeds in the standard way as follows:

$$h_{\ell,:} = W_{\ell,:,:} x_{\ell-1,:} + b_{\ell,:} \quad x_{\ell,:} = \phi(h_{\ell,:}), \qquad \text{for } \ell = 1, \ldots, D.$$

We will study the process arising from fixing an arbitrary input $x_{0,:} \in \{-1, 1\}^N$ and choosing the parameters independently at random: the entries of $W$ are sampled from $\text{Gauss}\left(0, \frac{\sigma_w^2}{N}\right)$, and the entries of $b$ from $\text{Gauss}\left(0, \sigma_b^2\right)$. For each $\ell \in [D]$, define $q_\ell = \frac{1}{N} \sum_{i=1}^{N} h_{\ell,i}^2$.

Note that for all $\ell \geq 1$, all the components of $h_{\ell,:}$ and $x_{\ell,:}$ are identically distributed.

### 2.3 THE WIDE-NETWORK LIMIT

For the purpose of defining a limit, assume that, for a fixed, arbitrary function $\chi : \mathbf{N} \to \{-1, 1\}$, for finite $N$, we have $x_{0,:} = (\chi(1), ..., \chi(N))$. For $\ell > 0$, if the limit exists (in the sense of "convergence in distribution"), let $\underline{x}_\ell$ be a random variable whose distribution is the limit of the distribution of $x_{\ell,1}$ as $N$ goes to infinity. Define $\underline{h}_\ell$ and $\underline{q}_\ell$ similarly.

### 2.4 TOTAL VARIATION DISTANCE

If $P$ and $Q$ are probability distributions, then $d_{TV}(P, Q) = \sup_E P(E) - Q(E)$, and if $p$ and $q$ are their densities, $d_{TV}(P, Q) = \frac{1}{2} \int |p(x) - q(x)| \, dx$.

## 3 CONVERGENCE IN PROBABILITY

In this section we characterize the length map of the hidden nodes of a deep network, for all activation functions satisfying the following assumptions.

**Definition 1** *An activation function $\phi$ is* permissible *if, (a) the restriction of $\phi$ to any finite interval is bounded; (b) $|\phi(x)| = \exp(o(x^2))$ as $|x|$ gets large.[1]; and (c) $\phi$ is measurable.*

Conditions (b) and (c) ensure that a key integral can be computed. The proof of Lemma 1 is in Appendix A.

---

[1] This condition may be expanded as follows, $\limsup_{x \to \infty} \frac{\log |\phi(x)|}{x^2} = 0$ and $\limsup_{x \to -\infty} \frac{\log |\phi(x)|}{x^2} = 0$.

**Lemma 1** *If $\phi$ is permissible, then, for all positive constants $c$, the function $g$ defined by $g(x) = \phi(cx)^2 \exp(-x^2/2)$ is integrable.*

Now, we recall the definition of a length map from [14]; we will prove that the the length process converges to this length map. Define $\tilde{q}_0, ..., \tilde{q}_D$ and $\tilde{r}_0, ..., \tilde{r}_D$ recursively as follows. First $\tilde{q}_0 = \tilde{r}_0 = 1$. Then, for $\ell > 0$,

$$\tilde{q}_\ell = \sigma_w^2 \tilde{r}_{\ell-1} + \sigma_b^2$$

and

$$\tilde{r}_\ell = \mathbb{E}_{z \in \text{Gauss}(0,1)}[\phi(\sqrt{\tilde{q}_\ell} z)^2].$$

If $\phi$ is permissible, then, since $\phi(cz)^2 \exp(-z^2/2)$ is integrable for all $c$, we have that $\tilde{q}_0, ..., \tilde{q}_D, \tilde{r}_0, ..., \tilde{r}_D$ are well-defined finite real numbers.

The following theorem shows that the length map $q_0, ..., q_D$ converges in probability to $\tilde{q}_0, ..., \tilde{q}_D$.

**Theorem 2** *For any permissible $\phi$, $\sigma_w, \sigma_b \geq 0$, any depth $D$, and any $\epsilon, \delta > 0$, there is an $N_0$ such that, for all $N \geq N_0$, with probability $1 - \delta$, for all $\ell \in \{0, ..., D\}$, we have $|q_\ell - \tilde{q}_\ell| \leq \epsilon$.*

The rest of this section is devoted to proving Theorem 2. Our proof will use the weak law of large numbers.

**Lemma 3 ([3])** *For any random variable $X$ with a finite expectation, and any $\epsilon, \delta > 0$, there is an $N_0$ such that, for all $N \geq N_0$, if $X_1, ..., X_N$ are i.i.d. with the same distribution as $X$, then*

$$\Pr\left( \left| \mathbb{E}[X] - \frac{1}{N} \sum_{i=1}^N X_i \right| > \epsilon \right) \leq \delta.$$

In order to divide our analysis into cases, we need the following lemma, whose proof is in Appendix B.

**Lemma 4** *If $\phi$ is permissible and not zero a.e., for all $\sigma_w > 0$, for all $\ell \in \{0, ..., D\}$, $\tilde{q}_\ell > 0$ and $\tilde{r}_\ell > 0$.*

We will also need a lemma that shows that small changes in $\sigma$ lead to small changes in $\text{Gauss}(0, \sigma^2)$.

**Lemma 5 (see [8])** *There is an absolute constant $C$ such that, for all $\sigma_1, \sigma_2 > 0$, $d_{TV}(\text{Gauss}(0, \sigma_1^2), \text{Gauss}(0, \sigma_2^2)) \leq C\frac{|\sigma_1 - \sigma_2|}{\sigma_1}$.*

The following technical lemma is proved in Appendix C.

**Lemma 6** *If $\phi$ is permissible, for all $0 < r \leq s$, for all $\beta > 0$, there is an $a \geq 0$ such that, for all $q \in [r, s]$, $\int_a^\infty \phi(\sqrt{q}z)^2 \exp(-z^2/2) \, dz \leq \beta$ and $\int_{-\infty}^{-a} \phi(\sqrt{q}z)^2 \exp(-z^2/2) \, dz \leq \beta$.*

Armed with these lemmas, we are ready to prove Theorem 2.

First, if $\phi$ is zero a.e., or if $\sigma_w = 0$, Theorem 2 follows directly from Lemma 3, together with a union bound over the layers. Assume for the rest of the proof that $\phi(x)$ is not zero a.e., and that $\sigma_w > 0$, so that $\tilde{q}_\ell > 0$ and $\tilde{r}_\ell > 0$ for all $\ell$.

For each $\ell \in [D]$, define $r_\ell = \frac{1}{N} \sum_{i=1}^N x_{\ell,i}^2$.

Our proof of Theorem 2 is by induction. The inductive hypothesis is that, for any $\epsilon, \delta > 0$ there is an $N_0$ such that, if $N \geq N_0$, then, with probability $1 - \delta$, for all $\ell' \leq \ell$, $|q_{\ell'} - \tilde{q}_{\ell'}| \leq \epsilon$ and $|r_{\ell'} - \tilde{r}_{\ell'}| \leq \epsilon$.

The base case holds because $q_0 = \tilde{q}_0 = r_0 = \tilde{r}_0 = 1$, no matter what the value of $N$ is.

Now for the induction step; choose $\ell > 0$, $0 < \epsilon < \min\{\tilde{q}_\ell/4, \tilde{r}_\ell\}$ and $0 < \delta \leq 1/2$. (Note that these choices are without loss of generality.) Let $\epsilon' \in (0, \epsilon)$ take a value that will be described later, using quantities from the analysis. By the inductive hypothesis, whatever the value of $\epsilon'$, there is an $N_0'$ such that, if $N \geq N_0'$, then, with probability $1 - \delta/2$, for all $\ell' \leq \ell - 1$, we have $|q_{\ell'} - \tilde{q}_{\ell'}| \leq \epsilon'$ and $|r_{\ell'} - \tilde{r}_{\ell'}| \leq \epsilon'$. Thus, to establish the inductive step, it suffices to show that, after conditioning

on the random choices before the $\ell$th layer, if $|q_{\ell-1} - \tilde{q}_{\ell-1}| \leq \epsilon'$, and $|r_{\ell-1} - \tilde{r}_{\ell-1}| \leq \epsilon'$, there is an $N_\ell$ such that, if $N \geq N_\ell$, then with probability at least $1 - \delta/2$ with respect only to the random choices of $W_{\ell,:,:}$ and $b_{\ell,:,:}$ that $|q_\ell - \tilde{q}_\ell| \leq \epsilon$ and $|r_\ell - \tilde{r}_\ell| \leq \epsilon$. Given such an $N_\ell$, the inductive step can be satisfied by letting $N_0$ be the maximum of $N_0'$ and $N_\ell$.

Let us do that. For the rest of the proof of the inductive step, let us condition on outcomes of the layers before layer $\ell$, and reason about the randomness only in the $\ell$th layer. Let us further assume that $|q_{\ell-1} - \tilde{q}_{\ell-1}| \leq \epsilon'$ and $|r_{\ell-1} - \tilde{r}_{\ell-1}| \leq \epsilon'$.

Recall that $q_\ell = \frac{1}{N} \sum_{i=1}^{N} h_{\ell,i}^2$. Since we have conditioned on the values of $h_{\ell-1,1}, ..., h_{\ell-1,N}$, each component of $h_{\ell,i}$ is obtained by taking the dot-product of $x_{\ell-1,:} = \phi(h_{\ell-1,:})$ with $W_{\ell,i,:}$ and adding an independent $b_{\ell,i}$. Thus, conditioned on $h_{\ell-1,1}, ..., h_{\ell-1,N}$, we have that $h_{\ell,1}, ..., h_{\ell,N}$ are independent. Also, since $x_{\ell-1,:}$ is fixed by conditioning, each $h_{\ell,i}$ has an identical Gaussian distribution.

Since each component of $W$ and $b$ has zero mean, each $h_{\ell,i}$ has zero mean.

Choose an arbitrary $i \in [N]$. Since $x_{\ell-1,:}$ is fixed by conditioning and $W_{\ell,i,1}, ..., W_{\ell,i,N}$ and $b_{\ell,i}$ are independent,

$$\mathbb{E}[q_\ell] = \mathbb{E}[h_{\ell,i}^2] = \sigma_b^2 + \frac{\sigma_w^2}{N} \sum_j x_{\ell-1,j}^2 = \sigma_b^2 + \sigma_w^2 r_{\ell-1} \stackrel{\text{def}}{=} \overline{q}_\ell. \tag{1}$$

We wish to emphasize the $\overline{q}_\ell$ is determined as a function of random outcomes before the $\ell$th layer, and thus a fixed, nonrandom quantity, regarding the randomization of the $\ell$th layer. By the inductive hypothesis, we have

$$|\mathbb{E}[q_\ell] - \tilde{q}_\ell| = |\mathbb{E}[h_{\ell,i}^2] - \tilde{q}_\ell| = |\overline{q}_\ell - \tilde{q}_\ell| = \sigma_w^2 |r_{\ell-1} - \tilde{r}_{\ell-1}| \leq \epsilon' \sigma_w^2. \tag{2}$$

The key consequence of this might be paraphrased by saying that, to establish the portion of the inductive step regarding $q_\ell$, it suffices for $q_\ell$ to be close to its mean. Now, we want to prove something similar for $r_\ell$. We have

$$\mathbb{E}[r_\ell] = \frac{1}{N} \sum_{i=1}^{N} \mathbb{E}[x_{\ell,i}^2] = \frac{1}{N} \sum_{i=1}^{N} \mathbb{E}[\phi(h_{\ell,i})^2] = \mathbb{E}[\phi(h_{\ell,1})^2],$$

since $h_{\ell,1}, ..., h_{\ell,N}$ are i.i.d. Recall that, earlier, we showed that $h_{\ell,i} \sim \text{Gauss}(0, \overline{q}_\ell)$. Thus

$$\mathbb{E}[r_\ell] = \mathbb{E}_{z \sim \text{Gauss}(0,\overline{q}_\ell)}[\phi(z)^2] = \mathbb{E}_{z \sim \text{Gauss}(0,1)}[\phi(\sqrt{\overline{q}_\ell}z)^2] = \sqrt{\frac{1}{2\pi}} \int \phi(\sqrt{\overline{q}_\ell}z)^2 \exp(-z^2/2) \, dz.$$

which gives

$$|\mathbb{E}[r_\ell] - \tilde{r}_\ell| \leq \left| \mathbb{E}_{z \sim \text{Gauss}(0,\overline{q}_\ell)}[\phi(z)^2] - \mathbb{E}_{z \sim \text{Gauss}(0,\tilde{q}_\ell)}[\phi(z)^2] \right|.$$

Since $|\overline{q}_\ell - \tilde{q}_\ell| \leq \epsilon' \sigma_w^2$ and we may choose $\epsilon'$ to ensure $\epsilon' \leq \frac{\tilde{q}_\ell}{2\sigma_w^2}$, we have $\tilde{q}_\ell/2 \leq \overline{q}_\ell \leq 2\tilde{q}_\ell$.

For $\beta > 0$ and $\kappa \in (0, 1/2)$ to be named later, by Lemma 6, we can choose $a$ such that, for all $q \in [\tilde{q}_\ell/2, 2\tilde{q}_\ell]$,

$$\int_{-\infty}^{-a} \phi(\sqrt{q}z)^2 \exp(-z^2/2) \, dz \leq \beta/2 \quad \text{and} \quad \int_{a}^{\infty} \phi(\sqrt{q}z)^2 \exp(-z^2/2) \, dz \leq \beta/2$$

and $\frac{1}{\sqrt{2\pi q}} \int_{-a}^{a} \exp\left(-\frac{z^2}{2q}\right) \, dz \geq 1 - \kappa$. Choose such an $a$.

We claim that $\left| \int_{-a}^{a} \phi(\sqrt{q}z)^2 \exp(-z^2/2)\, dz - \int \phi(\sqrt{q}z)^2 \exp(-z^2/2)\, dz \right| \leq \beta$ for all $\tilde{q}_\ell/2 < q \leq 2\tilde{q}_\ell$. Choose such a $q$. We have

$$\left| \int_{-a}^{a} \phi(\sqrt{q}z)^2 \exp(-z^2/2)\, dz - \int \phi(\sqrt{q}z)^2 \exp(-z^2/2)\, dz \right|$$

$$= \int_{-\infty}^{-a} \phi(\sqrt{q}z)^2 \exp(-z^2/2)\, dz + \int_{a}^{\infty} \phi(\sqrt{q}z)^2 \exp(-z^2/2)\, dz$$

$$\leq 2 \max \left\{ \int_{-\infty}^{-a} \phi(\sqrt{q}z)^2 \exp(-z^2/2)\, dz, \int_{a}^{\infty} \phi(\sqrt{q}z)^2 \exp(-z^2/2)\, dz \right\}$$

$$\leq \beta.$$

So now we are trying to bound $\left| \int_{-a}^{a} \phi(\sqrt{\overline{q}_\ell}z)^2 \exp(-z^2/2)\, dz - \int_{-a}^{a} \phi(\sqrt{\tilde{q}_\ell}z)^2 \exp(-z^2/2)\, dz \right|$ using $\tilde{q}_\ell/2 \leq \overline{q}_\ell \leq 2\tilde{q}_\ell$.

Using changes of variables, we have

$$\left| \int_{-a}^{a} \phi(\sqrt{\overline{q}_\ell}z)^2 \exp(-z^2/2)\, dz - \int_{-a}^{a} \phi(\sqrt{\tilde{q}_\ell}z)^2 \exp(-z^2/2)\, dz \right|$$

$$= \left| \frac{1}{\sqrt{\overline{q}_\ell}} \int_{-a\sqrt{\overline{q}_\ell}}^{a\sqrt{\overline{q}_\ell}} \phi(z)^2 \exp\left(-\frac{z^2}{2\overline{q}_\ell}\right) dz - \frac{1}{\sqrt{\tilde{q}_\ell}} \int_{-a\sqrt{\tilde{q}_\ell}}^{a\sqrt{\tilde{q}_\ell}} \phi(z)^2 \exp\left(-\frac{z^2}{2\tilde{q}_\ell}\right) dz \right|.$$

Since $\phi$ is permissible, $\phi^2$ is bounded on $[-a\sqrt{2\tilde{q}_\ell}, a\sqrt{2\tilde{q}_\ell}]$. If $P$ is the distribution obtained by conditioning $\text{Gauss}(0, \overline{q}_\ell)$ on $[-a\sqrt{\overline{q}_\ell}, a\sqrt{\overline{q}_\ell}]$, and $\tilde{P}$ by conditioning $\text{Gauss}(0, \tilde{q}_\ell)$ on $[-a\sqrt{\tilde{q}_\ell}, a\sqrt{\tilde{q}_\ell}]$, then if $M = \sqrt{2\pi} \sup_{z \in [-a\sqrt{2\tilde{q}_\ell}, a\sqrt{2\tilde{q}_\ell}]} \phi(z)^2$, since $\overline{q}_\ell \leq 2\tilde{q}_\ell$,

$$\left| \frac{1}{\sqrt{\overline{q}_\ell}} \int_{-a\sqrt{\overline{q}_\ell}}^{a\sqrt{\overline{q}_\ell}} \phi(z)^2 \exp(-\frac{z^2}{2\overline{q}_\ell})\, dz - \frac{1}{\sqrt{\tilde{q}_\ell}} \int_{-a\sqrt{\tilde{q}_\ell}}^{a\sqrt{\tilde{q}_\ell}} \phi(z)^2 \exp(-\frac{z^2}{2\tilde{q}_\ell})\, dz \right| \leq M d_{TV}(P, \tilde{P}).$$

But since, for $\kappa < 1/2$, conditioning on an event of probability at least $1 - \kappa$ only changes a distribution by total variation distance at most $2\kappa$, and therefore, applying Lemma 5 along with the fact that $|\overline{q}_\ell - \tilde{q}_\ell| \leq \epsilon' \sigma_w^2$, for the constant $C$ from Lemma 5, we get

$$d_{TV}(P, \tilde{P}) \leq 4\kappa + d_{TV}(\text{Gauss}(0, \overline{q}_\ell), \text{Gauss}(0, \tilde{q}_\ell))$$

$$\leq 4\kappa + \frac{C|\sqrt{\overline{q}_\ell} - \sqrt{\tilde{q}_\ell}|}{\sqrt{\tilde{q}_\ell}}$$

$$= 4\kappa + \frac{C|\overline{q}_\ell - \tilde{q}_\ell|}{|\sqrt{\overline{q}_\ell} + \sqrt{\tilde{q}_\ell}|\sqrt{\tilde{q}_\ell}}$$

$$\leq 4\kappa + \frac{C\epsilon'\sigma_w^2}{\tilde{q}_\ell}.$$

Tracing back, we have

$$\left| \int_{-a}^{a} \phi(\sqrt{\overline{q}_\ell}z)^2 \exp(-z^2/2)\, dz - \int_{-a}^{a} \phi(\sqrt{\tilde{q}_\ell}z)^2 \exp(-z^2/2)\, dz \right| \leq M \left( 4\kappa + \frac{C\epsilon'\sigma_w^2}{\tilde{q}_\ell} \right)$$

which implies

$$|\mathbb{E}[r_\ell] - \tilde{r}_\ell| \leq \left| \int \phi(\sqrt{\overline{q}_\ell}z)^2 \exp(-z^2/2)\, dz - \int \phi(\sqrt{\tilde{q}_\ell}z)^2 \exp(-z^2/2)\, dz \right|$$

$$\leq M \left( 4\kappa + \frac{C\epsilon'\sigma_w^2}{\tilde{q}_\ell} \right) + 2\beta.$$

If $\kappa = \min\{\frac{\epsilon}{24M}, \frac{1}{3}\}$, $\beta = \frac{\epsilon}{12}$, and $\epsilon' = \min\left\{ \frac{\epsilon}{2}, \frac{\epsilon}{2\sigma_w^2}, \frac{\tilde{q}_\ell}{2\sigma_w^2}, \frac{\tilde{q}_\ell\epsilon}{6CM\sigma_w^2} \right\}$ this implies $|\mathbb{E}[r_\ell] - \tilde{r}_\ell| \leq \epsilon/2$.

Recall that $q_\ell$ is an average of $N$ identically distributed random variables with a mean between $0$ and $2\tilde{q}_\ell$ (which is therefore finite) and $r_\ell$ is an average of $N$ identically distributed random variables, each with mean between $0$ and $\tilde{r}_\ell + \epsilon/2 \leq 2\tilde{r}_\ell$. Applying the weak law of large numbers (Lemma 3), there is an $N_\ell$ such that, if $N \geq N_\ell$, with probability at least $1 - \delta/2$, both $|q_\ell - \mathbb{E}[q_\ell]| \leq \epsilon/2$ and $|r_\ell - \mathbb{E}[r_\ell]| \leq \epsilon/2$ hold, which in turn implies $|q_\ell - \tilde{q}_\ell| \leq \epsilon$ and $|r_\ell - \tilde{r}_\ell| \leq \epsilon$, completing the proof of the inductive step, and therefore the proof of Theorem 2.

## 4 Diversity of behavior in the distribution of hidden nodes

In this section, we show that, for some activation functions, the probability distribution of hidden nodes can have some surprising properties.

### 4.1 Non-Gaussian

In this subsection, we will show that the hidden variables are sometimes not Gaussian. Our proof will refer to the Cauchy distribution.

**Definition 2** *A distribution over the reals that, for $x_0 \in \mathbf{R}$ and $\gamma > 0$, has a density $f$ given by $f(x) = \frac{1}{\pi\gamma\left[1+\left(\frac{x-x_0}{\gamma}\right)^2\right]}$ is a* Cauchy distribution*, denoted by* $\mathrm{Cauchy}(x_0, \gamma)$*.* $\mathrm{Cauchy}(0,1)$ *is the* standard Cauchy distribution.

**Lemma 7 ([6])** *If $X_1, ..., X_n$ are i.i.d. random variables with a Cauchy distribution, then $\frac{1}{n}\sum_{i=1}^n X_i$ has the same distribution.*

**Lemma 8 ([11])** *If $U$ and $V$ are zero-mean normally distributed random variables with the same variance, then $U/V$ has the standard Cauchy distribution.*

The following shows that there is a $\phi$ such that the limiting $h_2$ is not defined. It contradicts claims made on line 7 of Section A.1 of [14] and line 7 of Section 2.2 of [10].

**Proposition 9** *There is a $\phi$ such that, for every $\sigma_w > 0$, if $\sigma_b = 0$, then (a) for finite $N$, $h_{2,1}$ does not have a Gaussian distribution, and (b) $h_{2,1}$ diverges as $N$ goes to infinity.*

**Proof**: Consider $\phi$ defined by $\phi(y) = \begin{cases} 1/y & \text{if } y \neq 0 \\ 0 & \text{if } y = 0. \end{cases}$

Fix a value of $N$ and $\sigma_w > 0$, and take $\sigma_b = 0$. Each component of $h_{1,:}$ is a sum of zero-mean Gaussians with variance $\sigma_w^2/N$; thus, for all $i$, $h_{1,i} \sim \mathrm{Gauss}(0, \sigma_w^2)$. Now, almost surely, $h_{2,1} = \sum_{j=1}^N W_{2,1,j}\phi(h_{1,j}) = \sum_{j=1}^N W_{2,1,j}/h_{1,j}$. By Lemma 8, for each $j$, $W_{2,1,j}/h_{1,j}$ has a Cauchy distribution, and since $(NW_{2,1,1}), ..., (NW_{2,1,N}) \sim \mathrm{Gauss}(0, N\sigma_w^2)$, recalling that $h_{1,1}, ..., h_{1,N} \sim \mathrm{Gauss}(0, \sigma_w^2)$, we have that $NW_{2,1,1}/h_{1,1}, ..., NW_{2,1,N}/h_N^1$ are i.i.d. $\mathrm{Cauchy}(0, \sqrt{N})$. Applying Lemma 7, $h_{2,1} = \sum_{j=1}^N W_{2,1,j}\phi(h_{2,j}) = \frac{1}{N}\sum_{j=1}^N NW_{2,1,j}\phi(h_{1,j})$ is also $\mathrm{Cauchy}(0, \sqrt{N})$.

So, for all $N$, $h_{2,1}$ is $\mathrm{Cauchy}(0, \sqrt{N})$. Suppose that $h_{2,1}$ converged in distribution to some distribution $P$. Since the cdf of $P$ can have at most countably many discontinuities, we can cover the real line by a countable set of finite-length intervals $[a_1, b_1], [a_2, b_2], ...$ whose endpoints are points of continuity for $P$. Since $\mathrm{Cauchy}(0, \sqrt{N})$ converges to $P$ in distribution, for any $i$, $P([a_i, b_i]) \leq \lim_{N\to\infty} \frac{|b_i - a_i|}{\pi\sqrt{N}} = 0$. Thus, the probability assigned by $P$ to the entire real line is $0$, a contradiction. $\square$

### 4.2 Independence

The following contradicts a claim made on line 8 of Section A.1 of [14].

**Theorem 10** *If $\phi$ is either the ReLU or the Heaviside function, then, for every $\sigma_w > 0$, $\sigma_b \geq 0$, and $N \geq 2$, $(h_{2,1}, ..., h_{2,N})$ are not independent.*

**Proof**: We will show that $\mathbb{E}[h_{2,1}^2 h_{2,2}^2] \neq \mathbb{E}[h_{2,1}^2]\mathbb{E}[h_{2,2}^2]$, which will imply that $h_{2,1}$ and $h_{2,2}$ are not independent.

As mentioned earlier, because each component of $h_{1,:}$ is the dot product of $x_{0,:}$ with an independent row of $W_{1,:,:}$ plus an independent component of $b_{1,:}$, the components of $h_{1,:}$ are independent, and since $x_{1,:} = \phi(h_{1,:})$, this implies that the components of $x_{1,:}$ are independent. Since each row of $W_{1,:,:}$ and each component of the bias vector has the same distribution, $x_{1,:}$ is i.i.d.

We have

$$\mathbb{E}[h_{2,1}^2] = \mathbb{E}\left[\left[\left(\sum_{i\in[N]} W_{2,1,i}x_{1,i}\right) + b_{2,1}\right]^2\right]$$

$$= \sum_{(i,j)\in[N]^2} \mathbb{E}\left[W_{2,1,i}W_{2,1,j}x_{1,i}x_{1,j}\right] + \sum_{i\in[N]} \mathbb{E}\left[W_{2,1,i}x_{1,i}b_{2,1}\right] + \mathbb{E}\left[b_{2,1}^2\right].$$

The components of $W_{2,:,:}$ and $x_{1,:}$, along with $b_{2,1}$, are mutually independent, so terms in the double sum with $i \neq j$ have zero expectation, and $\mathbb{E}[h_{2,1}^2] = \left(\sum_{i\in[N]} \mathbb{E}\left[W_{2,1,i}^2\right] \mathbb{E}\left[x_{1,i}^2\right]\right) + \mathbb{E}[b_{2,1}^2]$. For a random variable $x$ with the same distribution as the components of $x_{1,:}$, this implies

$$\mathbb{E}[h_{2,1}^2] = \sigma_w^2 \mathbb{E}\left[x^2\right] + \sigma_b^2. \tag{3}$$

Similarly,
$\mathbb{E}[h_{2,1}^2 h_{2,2}^2]$

$$= \mathbb{E}\left[\left[\sum_{i\in[N]} W_{2,1,i}x_{1,i} + b_{2,1}\right]^2 \left[\sum_{i\in[N]} W_{2,2,i}x_{1,i} + b_{2,2}\right]^2\right]$$

$$= \sum_{(i,j,r,s)\in[N]^4} \mathbb{E}[W_{2,1,i}W_{2,1,j}W_{2,2,r}W_{2,2,s}x_{1,i}x_{1,j}x_{1,r}x_{1,s}]$$

$$+ 2\sum_{(i,j,r)\in[N]^3} \mathbb{E}[W_{2,1,i}W_{2,1,j}W_{2,2,r}x_{1,i}x_{1,j}x_{1,r}b_{2,2}] + 2\sum_{(i,r,s)\in[N]^3} \mathbb{E}[W_{2,1,i}W_{2,2,r}W_{2,2,s}x_{1,i}x_{1,r}x_{1,s}b_{2,1}]$$

$$+ 4\sum_{(i,r)\in[N]^2} \mathbb{E}[W_{2,1,i}W_{2,2,r}x_{1,i}x_{1,r}b_{2,1}b_{2,2}]$$

$$+ \sum_{(i,j)\in[N]^2} \mathbb{E}[W_{2,1,i}W_{2,1,j}x_{1,i}x_{1,j}b_{2,2}^2] + \sum_{(r,s)\in[N]^2} \mathbb{E}[W_{2,2,r}W_{2,2,s}x_{1,r}x_{1,s}b_{2,1}^2]$$

$$+ 2\sum_{i\in[N]} \mathbb{E}[W_{2,1,i}x_{1,i}b_{2,1}b_{2,2}^2] + 2\sum_{r\in[N]} \mathbb{E}[W_{2,2,r}x_{1,r}b_{2,1}^2b_{2,2}]$$

$$+ \mathbb{E}[b_{2,1}^2 b_{2,2}^2]$$

$$= \sum_{(i,r)\in[N]^2, i\neq r} \mathbb{E}[W_{2,1,i}^2 W_{2,2,r}^2]\mathbb{E}[x_{1,i}^2]\mathbb{E}[x_{1,r}^2] + \sum_{i\in[N]} \mathbb{E}[W_{2,1,i}^2 W_{2,2,i}^2]\mathbb{E}[x_{1,i}^4]$$

$$+ \sum_{i\in[N]} \mathbb{E}[W_{2,1,i}^2]\mathbb{E}[x_{1,i}^2]\mathbb{E}[b_{2,2}^2] + \sum_{r\in[N]} \mathbb{E}[W_{2,2,r}^2]\mathbb{E}[x_{1,r}^2]\mathbb{E}[b_{2,1}^2]$$

$$+ \mathbb{E}[b_{1,2}^2 b_{2,2}^2]$$

$$= \frac{(N^2 - N)\sigma_w^4 \mathbb{E}[x^2]^2}{N^2} + \frac{N\sigma_w^4 \mathbb{E}[x^4]}{N^2} + \frac{2N\sigma_w^2 \mathbb{E}[x^2]\sigma_b^2}{N} + \sigma_b^4$$

$$= \sigma_w^4 \mathbb{E}[x^2]^2 + \frac{\sigma_w^4(\mathbb{E}[x^4] - \mathbb{E}[x^2]^2)}{N} + 2\sigma_w^2 \sigma_b^2 \mathbb{E}[x^2] + \sigma_b^4.$$

Putting this together with (3), we have

$$\mathbb{E}[h_{2,1}^2 h_{2,2}^2] - \mathbb{E}[h_{2,1}^2]\mathbb{E}[h_{2,2}^2] = \frac{\sigma_w^4(\mathbb{E}[x^4] - \mathbb{E}[x^2]^2)}{N}. \tag{4}$$

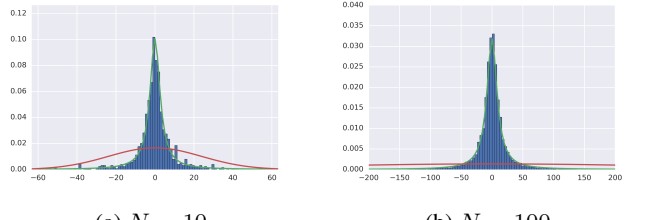

|                |                 |                 |
|----------------|-----------------|-----------------|
| (a) $N = 10$   | (b) $N = 100$   | (c) $N = 1000$  |

Figure 1: Histograms of $h[2, :]$, averaged over 100 random initializations, for $N \in \{10, 100, 1000\}$, along with $\mathrm{Cauchy}(0, \sqrt{N})$ (shown in green) and $\mathrm{Gauss}(0, \sigma^2)$ for $\sigma$ estimated from the data (shown in red).

Now, we calculate the difference using (4) for the Heaviside and ReLU functions.

**Heaviside.** Suppose $\phi$ is Heaviside function, i.e. $\phi(z)$ is the indicator function for $z > 0$. In this case, since the components of $h_{1,:}$ are symmetric about 0, the distribution of $x_{1,:}$ is uniform over $\{0, 1\}^N$. Thus $\mathbb{E}[x^4] = \mathbb{E}[x^2] = 1/2$, and so (4) gives $\mathbb{E}[h_{2,1}^2 h_{2,2}^2] - \mathbb{E}[h_{2,1}^2]\mathbb{E}[h_{2,2}^2] = \frac{3\sigma_w^4}{4N} \neq 0$.

**ReLU.** Next, we consider the case that $\phi$ is the ReLU. Recalling that, for all $i$, $h_{1,i} \sim \mathrm{Gauss}(0, \sigma_w^2)$, we have $\mathbb{E}[x^2] = \frac{1}{\sqrt{2\pi\sigma_w^2}} \int_0^\infty z^2 \exp\left(\frac{-z^2}{2\sigma_w^2}\right) dz$. By symmetry this is $\frac{1}{2}\mathbb{E}_{z\sim\mathrm{Gauss}(0,\sigma_w^2)}[z^2] = \sigma_w^2/2$. Similarly, $\mathbb{E}[x^4] = \frac{1}{2}\mathbb{E}_{z\sim\mathrm{Gauss}(0,\sigma_w^2)}[z^4] = \frac{3\sigma^4}{2}$. Plugging these into (4) we get that, in the case the $\phi$ is the ReLU, that

$$\mathbb{E}[h_{2,1}^2 h_{2,2}^2] - \mathbb{E}[h_{2,1}^2]\mathbb{E}[h_{2,2}^2] = \frac{\sigma_w^4\left((3/2)\sigma_w^4 - \sigma_w^4/4\right)}{N} = \frac{5\sigma_w^8}{4N} > 0,$$

completing the proof. $\qquad\square$

### 4.3 Undefined length map

Here, we show, informally, that for $\phi$ at the boundary of the second condition in the definition of permissibility, the recursive formula defining the length map $\tilde{q}_\ell$ breaks down. Roughly, this condition cannot be relaxed.

**Proposition 11** *For any $\alpha > 0$, if $\phi$ is defined by $\phi(x) = \exp(\alpha x^2)$, there exists a $\sigma_w, \sigma_b$ s.t. $\tilde{q}_\ell, \tilde{r}_\ell$ is undefined for all $\ell \geq 2$.*

**Proof**: Suppose $\sigma_w^2 + \sigma_b^2 = \frac{1}{4\alpha^2}$. Then $\tilde{q}_1 = \frac{1}{4\alpha^2}$, so that

$$\tilde{r}_1 = \frac{1}{\sqrt{2\pi}} \int_{-\infty}^\infty \phi(\sqrt{\tilde{q}_1}z) \exp\left(-\frac{z^2}{2}\right) dz = \frac{1}{\sqrt{2\pi}} \int_{-\infty}^\infty \exp(\alpha\sqrt{\tilde{q}_1}z^2) \exp\left(-\frac{z^2}{2}\right) dz$$

$$= \frac{1}{\sqrt{2\pi}} \int_{-\infty}^\infty \exp(z^2/2) \exp\left(-\frac{z^2}{2}\right) dz = \infty,$$

and downsteam values of $\tilde{q}_\ell$ and $\tilde{r}_\ell$ are undefined. $\qquad\square$

## 5 Experiments

Our first experiment fixed $x[0, :] = (1, ..., 1)$, $\sigma_w = 1$, $\sigma_b = 0$, $\phi(z) = 1/z$.

For each $N \in \{10, 100, 1000\}$, we (a) initialized the weights 100 times, (b) plotted the histograms of all of the values of $h[2, :]$, along with the $\mathrm{Cauchy}(0, \sqrt{N})$ distribution from the proof of Proposition 9, and $\mathrm{Gauss}(0, \sigma^2)$ for $\sigma$ estimated from the data. Consistent with the theory, the $\mathrm{Cauchy}(0, \sqrt{N})$ distribution fits the data well.

To illustrate the fact that the values in the second hidden layer are not independent, for $N = 1000$ and the parameters otherwise as in the other experiment, we plotted histograms of the values seen

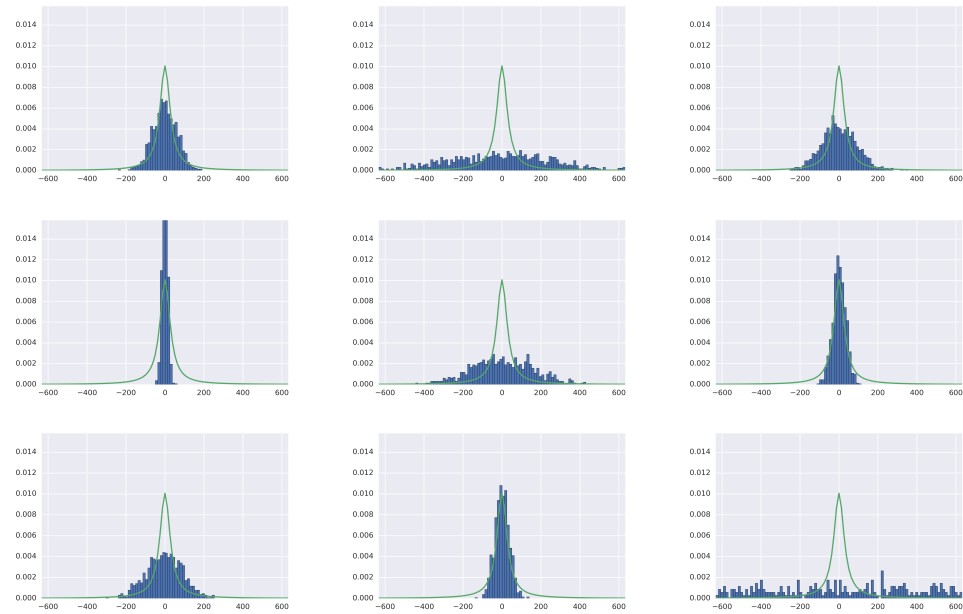

Figure 2: Histograms of $h[2, :]$ for nine random weight initializations.

in the second layer for nine random initializations of the weights in Figure 2. When some of the values in the first hidden layer have unusually small magnitude, then the values in the second hidden layer coordinately tend to be large. This is in contrast with the claim made at the end of Section 2.2 of [10]. Note that this is consistent with Theorem 2 establishing convergence in probability for permissible $\phi$, since the $\phi$ used in this experiment is not permissible.

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

## A PROOF OF LEMMA 1

Choose $c > 0$. Since $\limsup_{x \to \infty} \frac{\log |\phi(x)|}{x^2} = 0$ and $\limsup_{x \to -\infty} \frac{\log |\phi(x)|}{x^2} = 0$, we also have $\limsup_{x \to \infty} \frac{\log |\phi(cx)|}{x^2} = 0$ and $\limsup_{x \to -\infty} \frac{\log |\phi(cx)|}{x^2} = 0$. Thus, there is an $a$ such that, for all $x \notin [-a, a]$, $\log |\phi(cx)| \leq \frac{x^2}{8}$, which implies $\phi(cx)^2 \leq \exp\left(\frac{x^2}{4}\right)$. Since $\phi$ is permissible, it is bounded on $[-a, a]$. Thus, we have

$$\int \phi(cx)^2 \exp(-x^2/2) \, dx$$

$$= \int_{-\infty}^{-a} \phi(cx)^2 \exp(-x^2/2) dx + \int_{-a}^{a} \phi(cx)^2 \exp(-x^2/2) dx + \int_{a}^{\infty} \phi(cx)^2 \exp(-x^2/2) dx$$

$$\leq \int_{-\infty}^{-a} \exp(-x^2/4) dx + \left( \sup_{x \in [-a,a]} \phi(cx)^2 \right) \int_{-a}^{a} \exp(-x^2/2) dx + \int_{a}^{\infty} \exp(-x^2/4) dx$$

$$< \infty$$

completing the proof.

## B PROOF OF LEMMA 4

The proof is by induction. The base case holds since $\tilde{q}_0 = \tilde{r}_0 = 1$.

To prove the inductive step, we need the following lemma.

**Lemma 12** *If $\phi$ is not zero a.e., then, for all $c > 0$, $\mathbb{E}_{z \in \text{Gauss}(0,1)}(\phi(cz)^2) > 0$.*

**Proof**: If $\mu$ is the Lebesgue measure, since

$$\mu(\{x \in \mathbf{R} : \phi^2(cx) > 0\}) = \lim_{n \to \infty} \mu(\{x : \phi^2(cx) > 1/n\} \cap [-n, n]) > 0,$$

there exists $n$ such that $\mu(\{x : \phi^2(cx) > 1/n\} \cap [-n, n]) > 0$. For such an $n$, we have

$$\mathbb{E}_{z \in \text{Gauss}(0,1)}(\phi(cz)^2) \geq \frac{1}{n} e^{-n^2/2} \mu(\{x : \phi^2(cx) > 1/n\} \cap [-n, n]) > 0.$$

$\square$

Returning to the proof of Lemma 4, by the inductive hypothesis, $\tilde{r}_{\ell-1} > 0$, which, since $\sigma_w > 0$, implies $\tilde{q}_\ell > 0$. Applying Lemma 12 yields $\tilde{r}_\ell > 0$.

## C  PROOF OF LEMMA 6

Since $\limsup_{x \to \infty} \frac{\log |\phi(x)|}{x^2} = 0$ there is an $b$ such that, for all $x \geq b$, $\log |\phi(x)| \leq \frac{x^2}{8s}$, which implies $\phi(x)^2 \leq \exp\left(\frac{x^2}{4s}\right)$. Now, choose $q \in [r, s]$. For $a = b/\sqrt{r}$, we then have

$$
\int_a^\infty \phi(\sqrt{q}x)^2 \exp(-x^2/2) \; dx
$$

$$
= \frac{1}{\sqrt{q}} \int_{a\sqrt{q}}^\infty \phi(z)^2 \exp\left(-\frac{z^2}{2q}\right) \; dz
$$

$$
\leq \frac{1}{\sqrt{q}} \int_{a\sqrt{q}}^\infty \exp\left(\frac{z^2}{4s}\right) \exp\left(-\frac{z^2}{2q}\right) \; dz
$$

$$
\leq \frac{1}{\sqrt{q}} \int_{a\sqrt{q}}^\infty \exp\left(-\frac{z^2}{4q}\right) \; dz
$$

$$
\leq \frac{1}{\sqrt{q}} \int_b^\infty \exp\left(-\frac{z^2}{4q}\right) \; dz.
$$

By increasing $b$ if necessary, we can ensure $\frac{1}{\sqrt{q}} \int_b^\infty \exp\left(-\frac{z^2}{4q}\right) \; dz \leq \beta$ which then gives $\int_a^\infty \phi(\sqrt{q}x)^2 \exp(-x^2/2) \; dx \leq \beta$. A symmetric argument yields $\int_{-\infty}^a \phi(\sqrt{q}x)^2 \exp(-x^2/2) \; dx \leq \beta$, completing the proof.

