# OpenReview forum: "On the effect of the activation function on the distribution of hidden nodes in a deep network"
_ICLR.cc/2019/Conference_

### Official Review · AnonReviewer1 · 2018-10-31
**Technically correct, not well-written**

**Rating:** 4
**Confidence:** 3

**Review:**

Summary: the paper proves the convergence of empirical length map (length process) in NN to the length map for a permissible activation functions in a wide-network limit. The authors also show why the assumptions on the permissible functions can not be relaxed.

Quality: the paper seems to be technically correct. However, the authors do not discuss any consequence of their result. Why was it important to prove it? What does it tell us about the networks? While the proof may be of interest to the authors of [14] to correct their (possible) mistakes, I think the paper will go under the radar for most people and thus encourage the authors to heavily revise the paper.

Clarity: the writing is clear in general. The proofs sometimes jump over non-trivial things and explain easy steps, but that maybe subjective. The paper spends no effort explaining the contribution and its consequences.

Originality: the proven statements are novel and extend/fix the claims of [14]

Significance: as said above, I believe that in the current form the paper will have little to no impact. The importance of proving the main statement under more general conditions on activation functions is doubtful and the authors do not comment on that.

Minor comments:
* when introducing T{i,:,:} the <> notation is not clear. I could guess it from the later usage of the symbol, but these brackets can mean a lot of things, e.g. bracket mean (Section 2.1)
* it would be beneficial to define the main objects, wide-network limits, in a more formal way (Section 2.3)
* how the wide regime (large N) is interesting for studying deep NNs? [14] discusses that to some extent, but this should be explained here as well
* q_0 is never defined
* it's good practice to add numbers to all equations
* I believe the claim in the appendix of [14] was meant to be conditionally independent (see also the reviews of [14]). It's clear that preactivations should not be independent and, while technically interesting, spending a page of theorem 10 and on plots seems unnecessary. Even in the paper's example preactivations are uncorrelated in the limit of large N.
* I don't see the point of having experiments in this paper. The authors have already proven the fact. Also, it is not clear how to read the plots (no axes, little description) and come to the statements from page 9.

********************
After the authors' response:
If the main motivation of the paper is to fix the mistakes in [14], then the paper should clearly state so, in addition to explaining why fixing is necessary. While I believe that pointing out other paper's mistakes and correcting them is important, the current state of the paper leads me to keeping my initial score and recommending to reject the paper.

---

> ### Author Response · Authors · 2018-11-06
> **significance, and independence vs. near-independence**
>
> We thank this reviewer for the valuable detailed suggestions regarding presentation.  Thanks also for pointing out the need to define q_0 before the statement of Theorem 2;  q_0 = 1.
>
> Regarding motivation, as we pointed out to Reviewer 1, the length map studied in our paper, itself published in NIPS’16, has been repeatedly applied in a long series of papers published in NIPS, ICLR and ICML; we provide a list in the fourth paragraph of our paper.  It is therefore noteworthy that the logic supporting this length map published in [14] is not valid, and the claim made about it in that paper is incorrect.  This then motivates the question of what similar statement is correct.
>
> We have spoken to two of the authors of [14] about our findings, and neither of them claimed that they meant “conditionally independent” when they wrote “independent”.  While, on a first reading, this claim in their paper struck us as highly implausible, we felt that it was necessary to prove our claim that their claim was incorrect.   The experiments illustrate the strength of some of the effects analyzed in our paper.
>
> You are correct that, as N gets large, the dependence between pairs of preactivation values becomes weaker.  But then, when analyzing the next layer, there are competing effects: as N gets larger, the dependence between individual pairs of hidden nodes is approaching zero but the number of such interferences is approaching infinity, but the improved stability obtained by averaging more cases is improving.  A rigorous analysis must take account of all of these.

---

### Official Review · AnonReviewer3 · 2018-11-02
**In this paper, the authors studied how the activation function affects the behavior of randomized deep networks.**

**Rating:** 5
**Confidence:** 3

**Review:**

* summary
In this paper, the authors studied how the activation function affects the behavior of randomized deep networks.
When the activation function is permissible and the weights of DNN are generated from the Gaussian distribution,
the output of each layer was related to the so-called length process. When the permissibility is violated,
the convergence property may not hold. Some numerical experiments confirm the theoretical findings.


* comments
However, The randomized DNN is not clear whether theoretical results in this paper is related to the practical DNN.
The authors showed intensive proofs of theorems.
I think that the relation between DNN in practice and the results in this paper should be pursued more.

* The meaning of Theorem 10 is not clear. What does the theorem reveal about the ReLU function in the practical usage?

* In this paper, a limit theorem in terms of the dimension N is considered.
  However, the limit theorem in terms of the depth D is also important for the DNN.
  Some comments on that would be helpful for readers.

* Is there any relation between the analysis in this paper and batch normalization or weight normalization?

---

> ### Author Response · Authors · 2018-11-06
> **motivation, and discussion of normalization**
>
> Theorem 10 demonstrates a flaw in the logic provided in [14], motivating a new analysis.  As discussed in some earlier papers in this line of research, if sigma_w and sigma_b are chosen so that q_{ell} = q_{ell-1}, very deep networks can be trained.
>
> We agree that extending a rigorous analysis to concern batch normalization and weight normalization is an interesting direction for further research.  It seems that the most interesting effects would occur during training.  Framing this for tractable rigorous analysis looks like an interesting and important challenge.

---

### Official Review · AnonReviewer2 · 2018-11-04
**an abstract analysis that does not aim to derive any conclusions**

**Rating:** 4
**Confidence:** 3

**Review:**

This paper performs an analysis of the length scale of activations for deep fully-connected neural networks with respect to the activation function in neural networks. The authors show that for a very large class of activation functions, the length process converges in probability.

I am listing my main concerns about this manuscript below.

1. The paper is poorly motivated and does not make an attempt to relate its results to observations in practice or the design of new techniques. It is an abstract analysis of the probability distribution of the activations.

2. Theorem 2, which is the main theoretical contribution of the paper, hinges on fixing the inputs of the neural network with weights sampled randomly from a Gaussian distribution. It is difficult to connect this with practice. This is not unreasonable and indeed common in mean-field analyses. However such analyses go further in their implications, e.g., https://arxiv.org/abs/1606.05340, https://arxiv.org/abs/1806.05393 etc. This is my main concern about the paper, its lack of concrete implications despite the simplifying assumptions.

3. It would be very interesting if the analysis in this manuscript informs new activation functions or new initialization methods for training deep networks.

---

> ### Author Response · Authors · 2018-11-06
> **motivation for our analysis**
>
> The length map studied in our paper, itself published in NIPS’16, has been repeatedly applied in a long series of papers published in NIPS, ICLR and ICML; we provide a list in the fourth paragraph of our paper.  It is therefore noteworthy that the logic supporting this length map published in [14] is not valid, and the claim made about it in that paper is incorrect.  This in turn motivates the question of what similar statement is correct.

---

### Meta-Review · Area_Chair1 · 2018-12-11
**Too narrow**

**Confidence:** 4
**Recommendation:** Reject

**Metareview:**

I appreciate that the authors are refuting a technical claim in Poole et al., however the paper has garnered zero enthusiasm the way it is written. I suggest to the authors that they rewrite the paper as a refutation of Poole et al., and name it as such.